

# Comprehensive map and functional annotation of the mouse white adipose tissue proteome

Xiaoyue Tang[1,*], Juan Li[2,*], Wei-gang Zhao[2], Haidan Sun[1], Zhengguang Guo[1], Li Jing[1], Zhufang She[3], Tao Yuan[2], Shuai-nan Liu[3], Quan Liu[3], Yong Fu[2] and Wei Sun[1]

[1] Core Facility of Instrument, Institute of Basic Medical Sciences, Chinese Academy of Medical Sciences/School of Basic Medicine, Peking Union Medical College, Beijing, China
[2] Department of Endocrinology, Key Laboratory of Endocrinology of Ministry of Health, Peking Union Medical College Hospital, Chinese Academy of Medical Science and Peking Union Medical College, Beijing, China
[3] State Key Laboratory of Bioactive Substances and Functions of Natural Medicines, Institute of Materia Medica, Chinese Academy of Medical Sciences and Peking Union Medical College, Diabetes Research Center of Chinese Academy of Medical Sciences, Beijing, China
* These authors contributed equally to this work.

## ABSTRACT

White adipose tissue (WAT) plays a significant role in energy metabolism and the obesity epidemic. In this study, we sought to (1) profile the mouse WAT proteome with advanced 2DLC/MS/MS approach, (2) provide insight into WAT function based on protein functional annotation, and (3) predict potentially secreted proteins. A label-free 2DLC/MS/MS proteomic approach was used to identify the WAT proteome from female mouse WAT. A total of 6,039 proteins in WAT were identified, among which 5,160 were quantified (spanning a magnitude of $10^6$) using an intensity-based absolute quantification algorithm, and 3,117 proteins were reported by proteomics technology for the first time in WAT. To comprehensively analyze the function of WAT, the proteins were divided into three quantiles based on abundance and we found that proteins of different abundance performed different functions. High-abundance proteins (the top 90%, 1,219 proteins) were involved in energy metabolism; middle-abundance proteins (90–99%, 2,273 proteins) were involved in the regulation of protein synthesis; and low-abundance proteins (99–100%, 1,668 proteins) were associated with lipid metabolism and WAT beiging. Furthermore, 800 proteins were predicted by SignalP4.0 to have signal peptides, 265 proteins had never been reported, and five have been reported as adipokines. The above results provide a large dataset of the normal mouse WAT proteome, which might be useful for WAT function research.

# INTRODUCTION

Mammals have two types of adipose tissue, brown adipose tissue (BAT) and white adipose tissue (WAT). Brown adipose tissue dissipates energy and increases fatty acid oxidation

Corresponding authors
Wei-gang Zhao, zhaowg@pumch.cn
Wei Sun, sunwei@ibms.pumc.edu.cn

and heat production by uncoupled respiration, which is mediated by uncoupling protein-1 (UCP1) (*Klingenberg, 1999*; *Cannon & Nedergaard, 2004*). The primary function of WAT is to store energy in lipid droplets in the form of triacylglycerols (TAGs) (*Kuhnlein, 2012*). White adipose tissue also secrete hormones and cytokines to regulate energy intake, energy expenditure, and carbohydrate and lipid metabolism, e.g., adiponectin (ADI), leptin, and plasminogen activator inhibitor-1 (*McGown, Birerdinc & Younossi, 2014*). Furthermore, many studies have found that these hormones and cytokines are associated with diseases. For example, ADI, one of the most abundant adipokines produced by adipocytes, plays a fundamental role in maintaining vascular homeostasis, protecting from vascular injury and atherogenesis (*Vaiopoulos et al., 2012*; *Nishimura et al., 2008*). These proteins are considered a valuable resource in disease research. Therefore, a comprehensive profile of the WAT proteome will provide a basic understanding of WAT function and benefit disease studies.

The earliest study of the normal WAT proteome was from *Sanchez et al. (2001)*. In this study, 61 protein entries from mouse WAT were identified from one hundred and nine spots in a 2D-PAGE gel. Later, to optimize the resolution of WAT proteins, *Lanne et al. (2001)* used thiourea to obtain distinct spots in a 2D gel and finally identified 108 proteins from 140 spots in a 2D gel taken from the WAT of the epididymal fat pads of obese, diabetic ob/ob mice. *Hsieh et al. (2013)* analyzed proteins differentially expressed in the WAT of mice fed diets with or without bitter melon seed oil for 11 weeks using two-dimensional gel electrophoresis and finally identified 23 proteins from 30 spots. With the development of mass spectrometry technology, more proteins can be identified in WAT. In 2007, *Adachi et al. (2007)* investigated the adipocyte proteome by combining one-dimensional gel electrophoresis and on-line electrospray tandem mass spectrometry with biochemical procedures for the subfractionation of the 3T3-L1 adipocyte cellular proteome (nuclei, mitochondria, membrane, and cytosol). Finally, they reported 3,287 SwissProt annotated proteins in 3T3-L1 adipocytes. *Forner et al. (2009)* analyzed the mouse fat mitochondria proteome response to cold-induced thermogenesis. By combining HPLC and high-resolution quantitative linear ion trap (LTQ)-Orbitrap mass spectrometry with the SILAC method, they identified 2,434 proteins from the mitochondria of mouse adipose tissue. *Xie et al. (2010)* identified 1,493 adipocyte proteins isolated from abdominal adipose tissue using a one-dimensional SDS-PAGE and HPLC-ESI-MS/MS approach. *Feist et al. (2018)* analyzed mesenteric, omental, and uterine adipose tissue groups from the peritoneal cavities of young and aged C57BL/6J mouse cohorts with SDS-PAGE gel combination with LC–MS/MS method. They finally identified 2,308 protein groups and quantified 2,167 groups. There were also some studies to investigate differently expressed proteins in adipose tissue to unveil the mechanisms underlying adipose tissue dysfunction and understand the functional mechanism of WAT (*Peinado et al., 2011*; *Andrade et al., 2014*; *Kim et al., 2015*; *Ke et al., 2017*).

Presently, due to its high-throughput and high-sensitivity capacities, proteomics technology has garnered a lot of attention recently in biological research. The number of identified gene products in two proteome studies of human tissues and body fluids even

approached the number of protein-coding genes in the complete human genome (*Kim et al., 2014*; *Wilhelm et al., 2014*; *Consortium, 2012*).

Therefore, in this study, we sought to (1) profile the mouse WAT proteome with advanced 2DLC/MS/MS approach, (2) provide insight into WAT function based on protein functional annotation, and (3) predict potentially secreted proteins. In previous reports of WAT proteome, SDS-PAGE was the most commonly used method for protein separation, but its peak capacity was limit. 2DLC approach could greatly improve the peak capacity by the orthogonality of the two-dimensional separations; therefore, it could identify more proteins (*Gilar et al., 2005*; *Wang et al., 2011*; *Song et al., 2010*). Thus, 2DLC approach was used in our study. For mass spectrometer, Triple TOF 5600 with higher sensitivity and resolution was used in the present study. To this end, we performed an in-depth WAT proteomic analysis using normal mouse WAT samples. Qualitative and quantitative analysis were applied to comprehensively profile the mouse WAT proteome. The function of WAT and the potentially secreted protein prediction was annotated with bioinformatic analysis. Our data provide a comprehensive mouse WAT proteome dataset, which might be useful for WAT function research.

## EXPERIMENTAL PROCEDURES

### Animals

Six- to eight-week-old female C57BL/6J mice were purchased from HFK Bioscience Laboratories (Beijing, China). All mice were maintained with free access to water and food and maintained under SPF conditions and a 12 h light/dark cycle at 23 ± 2 °C. The mice were fed with a standard chow (10% lipids) diet for 22 weeks. All animals were handled according to the Standards for Laboratory Animals (GB14925-2001) and the Guideline on the Humane Treatment of Laboratory Animals (MOST 2006a) established by the People's Republic of China. The two guidelines were conducted in adherence to the regulations of the Ethics Committee of Peking Union Medical College Hospital Institutional Animal Care and Use Committee (IACUC) and all animal procedures were approved by the IACUC (approval number: SCXKBeijing-2009-0004). All efforts were made to minimize suffering.

### Preparation of WAT samples

A total of six C57BL/6J mice was applied to collect 300 mg gonadal WAT. Phosphate buffer solution was used to wash the tissue. The tissues were gently agitated by vortexing, and centrifuged at 1,000 g for 5 min. After the solution was removed. One mL lysis buffer (protein extraction kit, P1250-50, Applygen, Beijing) and one μL protease inhibitor cocktail (50x, P1265-0.5, Applygen, Beijing) (1:1,000) were added to the tissue. And the tissue was homogenized using a spinning blade tissue homogenizer (IKA R104, Janke & Kunkel KG.IKA-werk, Staufen, Germany) on ice for complete lysis. Two mL protein extraction regents were added to the tissue and kept in 4 °C for 40 min to separate lipid completely. After centrifugation at 4 °C for 10 min, the solution is divided into two phases and the middle of the two phases is protein membrane. The upper and lower two layers of liquid are removed and the protein membrane is attached to the centrifugal tube

wall. Protein dissolution was performed by adding a buffer solution containing seven M urea, two M thiourea, 65 mM DTE, and 83 mM Tris (Sigma-Aldrich, St. Louis, MO, USA). After 10 min of $20,000 \times g$ centrifugation at 4 °C, the supernatants were collected and stored at −80 °C. The Bradford assay with Bradford reagents (Thermo Fischer Scientific, Waltham, MA, USA) was supplied to measure the protein concentration of the WAT samples. The six WAT protein samples were pooled into one mixed sample with the same protein amount. The WAT proteins were prepared using filter-aided sample preparation (FASP) methods (*Wiśniewski et al., 2009*). The sample (100 μg in total) was deoxidized with 20 mM DTT and alkylated with 50 mM IAA. Then, the WAT proteins were loaded onto a 10 kDa filter device (micro Biospin, Nanosep 10K Omega; PALL, New York, NY, USA) and centrifuged at 14,000 g at 4 °C. After washing twice with UA (8 mol/L urea in 0.1 mol/L Tris–HCl, pH 8.5) and four times with 25 mmol/L $NH_4HCO_3$, the samples were digested with trypsin (Trypsin Gold, mass spec grade, Promega, WI, USA, enzyme to protein ratio of 1:50) and incubated at 37 °C overnight. After digestion, HLB 3 cc extraction cartridges (Oasis, Waters, Dublin, Ireland) were applied to desalt the peptides. Five hundred μL 0.1% formic acid cleaned the peptide three times and peptide was finally eluted with 500 μL 100% ACN. The pooled peptide elution was vacuum dried and stored at −80 °C.

## LC–MS/MS

The pooled peptide mixture was first fractioned using a high-pH reverse-phase liquid chromatography (hp-RPLC) column from Waters (4.6 mm × 250 mm, $C_{18}$, three μm). The sample was loaded onto the column in buffer A1 (pH = 10). Peptides were separated and eluted from 5% B1 to 90% B1 (90% ACN; pH = 10, flow rate, one mL/min) in 60 min. The eluted peptides were collected at a rate of one fraction per minute and we concatenated the 60 collected fractions into 20 fractions (concatenation scheme: 1 + 21 + 41, 2 + 22 + 42, etc). Each pooled sample was analyzed on a RP-$C_{18}$ self-packing LC column (75 μm × 100 mm, three μm). The eluted gradient was 5–30% buffer B2 (0.1% formic acid, 99.9% ACN; flow rate, 0.5 μL/min) for 60 min.

The eluted peptide was measured using LC–MS instrumentation consisting of a nano source to TripleTOF 5600 mass spectrometer. High sensitivity mode and the following parameters were used when acquiring the MS/MS data. MS data were acquired with 30 data-dependent MS/MS scans per every full scan and full scans were acquired at a resolution of 40,000 and MS/MS scans at 20,000. Thirty-five percent normalized collision energy and charge state screening (including precursors with +2 to +4 charge state) was applied. MS/MS scans were performed at the scan range of m/z 100–1,800 with the scan time of 100 ms. Dynamic exclusion was set to 15 s to avoid repeated sequencing of identical peptides.

## Data processing

The MS/MS spectra were processed using Mascot software (version 2.3.02, Matrix Science, London, UK) and searched against the SwissProt mouse database from the UniProt website (http://www.UniProt.org). The specificity of trypsin digestion was set for cleavage after K or R and two missed cleavage sites in the trypsin digestion were allowed. The fixed

modification was carbamidomethylation of cysteine in Mascot. Deamidation of asparagine and glutamine and the oxidation of methionine were specified as variable modifications. The parent ion tolerance was set to 10 ppm and the fragment ion mass tolerance was set to 0.05 Da. Mascot search results were further validated using Scaffold (version 4.3.2; Proteome Software Inc., Portland, OR, USA). Peptide identifications were accepted at an false discovery rate (FDR) less than 1.0% by the Scaffold Local FDR algorithm and were determined by searching a reverse database. Proteins that contained similar peptides and could not be differentiated based on MS/MS analysis alone were grouped to satisfy the principles of parsimony. Proteins sharing significant peptide evidence were grouped into clusters. Protein identifications were accepted if they could achieve a FDR of less than 1.0% and contain at least two uniquely identified peptides.

To rank the relative abundance of proteins, an intensity-based absolute quantification (iBAQ) algorithm was used (*Schwanhäusser et al., 2011*). The detailed protocol was provided below:

(1) The protein intensities were first computed by Progenesis LC–MS (v2.6, Nonlinear Dynamics, Newcastle upon Tyne, UK) as the sum of all identified peptide intensities (maximum peak intensities of the peptide elution profile, including all peaks in the isotope cluster).
(2) The protein intensities were then divided by the number of theoretically observable peptides (calculated by in silico protein digestion; all fully tryptic peptides between six and 30 amino acids were counted). The resulting intensities were iBAQ values, which are shown as "Absolute iBAQ intensities".
(3) The relative iBAQ intensities were computed by dividing the absolute iBAQ intensities by the sum of all absolute iBAQ intensities.
(4) The relative iBAQ intensities were applied to estimate the estimated protein abundances (the proportions of protein amounts to total WAT protein amount). The estimated protein abundances in WAT were finally calculated by multiplying the relative iBAQ intensities by theoretically relative molecular mass.
(5) The percentage was computed by dividing the estimated protein abundances by the sum of all estimated protein abundances.

## Functional annotation

Gene Ontology (GO) was performed using the PANTHER database (Protein Analysis Through Evolutionary Relationships, http://www.pantherdb.org/). Proteins were divided into cellular components, biological processes, and molecular functions based on their functional annotations. When more than one assignment was available, all the functional annotations were considered in the results.

The ingenuity pathway analysis (IPA, Ingenuity Systems, Mountain View, CA, USA) database was used to analyze canonical pathways. The SwissProt accession numbers of the identified proteins were uploaded into the IPA using default settings. This analysis was focused on the pathways and functions in which the proteins are expected to be involved.

All identified WAT proteins, high, middle, and low-abundance proteins were analyzed respectively.

All proteins were used for potentially secreted protein prediction. For this purpose, the SwissProt accession numbers were input into SignalP 4.1 (http://www.cbs.dtu.dk/services/SignalP/) (*Ke et al., 2017*). This software was used to predict the N-terminal signal peptides for classically secreted proteins.

## RESULTS

### Workflow of proteome analysis

In this study, samples of WAT from six C57BL/6J female mice were used to generate a comprehensive profile of the normal mouse WAT proteome. The extracted proteins were digested using the FASP method (*Wiśniewski et al., 2009*). The digested peptides were first separated by high-pH RPLC and mixed into 20 fractions and analyzed by nanoRPLC–MS/MS in three replicates. The peak intensity-based semiquantification method iBAQ was used to estimate the abundance of the quantified proteins. The PANTHER classification system and IPA were applied to annotate the function of WAT. The SignalP 4.1 Server was applied to predict secreted proteins (Fig. S1).

### A comprehensive profile of the WAT proteome

In this study, pooled WAT samples were used to establish a large database for the WAT proteome. All of 516,703 spectra, 60,215 unique peptides and 6,039 proteins were identified (FDR<1% at the protein and peptide level with at least two unique peptides for each protein) (Detailed data provided in Tables S1–S3). Among the 6,039 identified proteins, 77% had molecular weight between 10 and 90kD (Fig. S2). Five thousand and fifty-seven proteins (84%) of the 6,039 proteins were identified in the three replicates (Fig. 1A).

In order to improve our understanding of the WAT proteome and facilitate the development of WAT functional annotation, quantitation of WAT proteins was performed. Among the 6,039 identified proteins, 5,270 proteins could be quantified by the peak intensity-based semi-quantification method iBAQ (*Schwanhäusser et al., 2011*) (Fig. 1B). As for the correlation of protein quantification, the correlation of two replicates was approximately 0.98 (Fig. S3), indicating good LC/MS/MS reproducibility. The proteins with technical CVs greater than 0.3 were excluded from further analysis in order to reduce the interference of technical variation, and a total of 5,160 WAT proteins (Fig. 1C; Table S4) were kept. The relative protein abundance spans six orders of magnitude. We divided the quantified proteins into three quantiles based on protein abundance: high- (the top 90%, 1,219 proteins), middle- (90–99%, 2,273 proteins), and low-abundance proteins (99–100%, 1,668 proteins). Additionally, Table 1 listed top ten abundant proteins quantified with the iBAQ algorithm in our research. The ten proteins account for 20% of the total WAT protein abundance. Carbonic anhydrase 3, vimentin and polymerase I and transcript release factors are associated with fatty acid metabolism, adipogenesis, and lipolysis, respectively (*Lyons et al., 1991*; *Joo et al., 2010*; *Aboulaich et al., 2011*). Galectin-1 has been proposed as an early marker
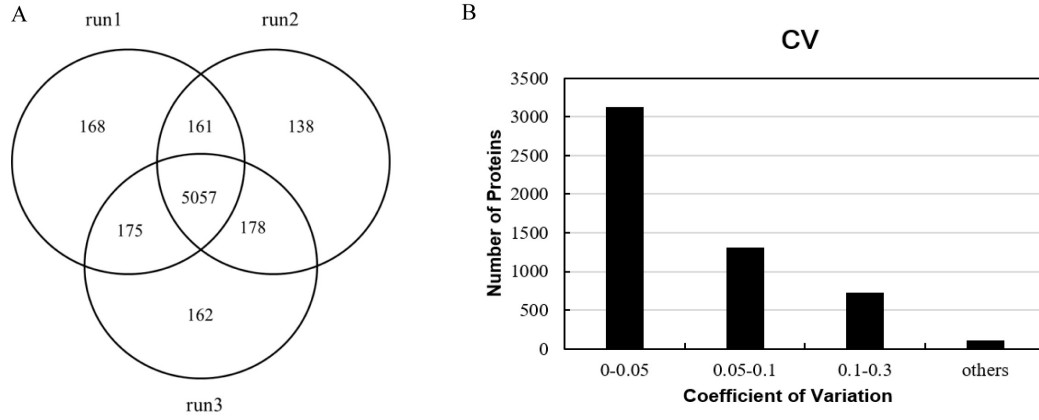

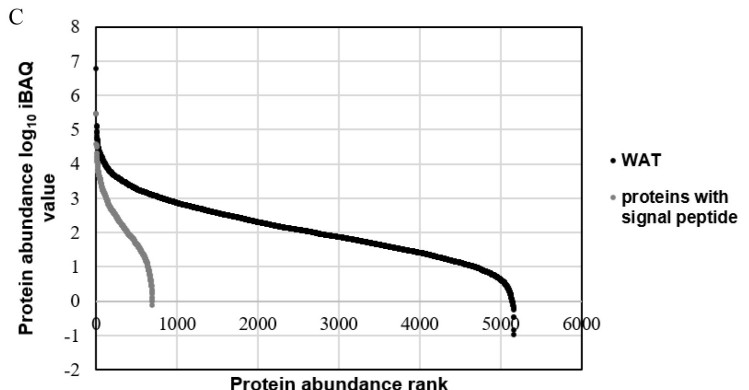

**Figure 1 Proteome profile analysis of the normal WAT proteome.** (A) Venn diagram of protein identification in three technical replicates. (B) Technical variation of the three replicates. (C) Quantitative protein abundance range in WAT samples with the iBAQ algorithm.

**Table 1 The top ten most abundant proteins in the mouse WAT proteome.**

| Accession number | Name | iBAQ value | Percentage (%) |
|---|---|---|---|
| P07724 | Serum albumin | 2.87E+05 | 9.33 |
| P16015 | Carbonic anhydrase 3 | 1.21E+05 | 1.68 |
| O54724 | Polymerase I and transcript release factor | 8.01E+04 | 1.67 |
| P58774 | Tropomyosin beta chain | 8.72E+04 | 1.36 |
| P20152 | Vimentin | 4.93E+04 | 1.25 |
| P31001 | Desmin | 4.23E+04 | 1.07 |
| Q60605 | Myosin light polypeptide 6 | 1.30E+05 | 1.04 |
| P37804 | Transgelin | 8.74E+04 | 0.93 |
| P16045 | Galectin-1 | 1.20E+05 | 0.85 |
| Q08091 | Calponin-1 | 5.05E+04 | 0.80 |

of adipocyte differentiation and to be actively involved in adipose tissue development (*Ahmed et al., 2010*; *Wang et al., 2004*). The expression of transgelin was associated with the development of white adipocyte, and thought to begin with progenitor cells from the
**Table 2 Proteomic studies on the mouse WAT proteome.**

| No. | Year | Protein identification | Fraction methods | MS | FDR | Reference |
|---|---|---|---|---|---|---|
| 1 | 2001 | 61 | 2D-PAGE | MALDI-TOF-MS | | Sanchez et al. (2001) |
| 2 | 2001 | 108 | 2D-SDS-PAGE | MALDI-TOF-MS, Q-TOF | | Lanne et al. (2001) |
| 3 | 2007 | 3,287 | SDS-PAGE followed by LC | LTQ-FTICR | | Adachi et al. (2007) |
| 4 | 2009 | 2,434 | SDS-PAGE followed by HPLC | LTQ-Orbitrap | Protein level FDR < 1% | Forner et al. (2009) |
| 5 | 2010 | 1,493 | SDS-PAGE followed by HPLC | LTQ-FTICR | Protein probability score ≥ 99% | Xie et al. (2010) |
| 6 | 2018 | 2,308 | Low-fraction SDS-PAGE gel followed by LC | Q-Exactive | Protein level FDR = 1% | Feist et al. (2018) |
| 7 | 2018 | 6,036 | RPLC | Triple TOF | Protein level FDR = 1% | This study |

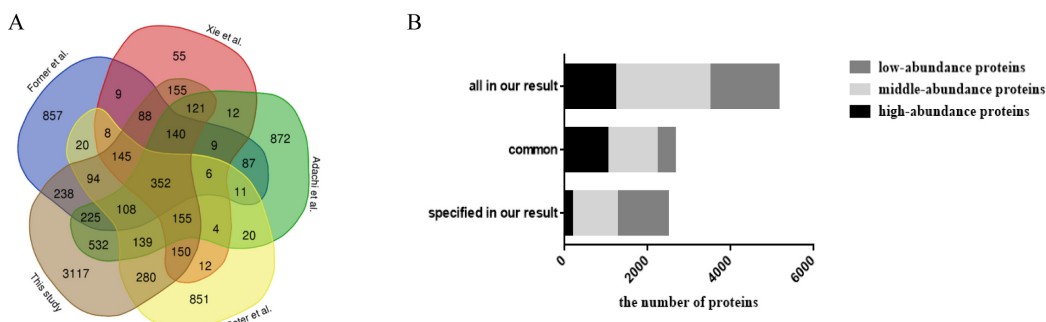

**Figure 2 Comparison analysis of the present mouse adipose tissue proteome and protein distribution analysis of WAT proteome in this study.** (A) Venn diagram analysis of present mouse WAT proteome studies. (B) Protein distribution based on abundance of WAT proteome in this study.

perivascular compartment (pericytes) in WAT (Todorčević et al., 2010). Other three proteins, tropomyosin (beta chain), desmin, and myosin light polypeptide 6, were cytoskeletal proteins.

When comparing with previous studies (Table 2; Fig. 2A), approximately 58% of the fat mitochondria proteome, 60% of the mesenteric, omental, and uterine adipose tissue proteome, 63% of the 3T3-L1 adipocyte proteome and 92% of the adipocyte proteome (isolated from abdominal adipose tissue) was overlapped with our study, respectively. When all data from these four studies were combined, 352 proteins were detected in all of the studies and 3,117 proteins from our results were reported for the first time by proteomics technology in WAT. Among common proteins (Fig. 2B), high- and middle-abundance proteins contributed about 84%. And among specific proteins, less (7%) high-abundance proteins were found in our study. When comparing previous study with our high-abundance proteins (Fig. S4), 177 proteins were firstly reported in our study and 85% (1,042 proteins) of our high-abundance proteins could be identified previously. The above data showed that most high abundance proteins (85%) and middle abundance proteins (52%) were identified by previous reports, and less proteins low abundant proteins (26%). The possible reasons might come from the proteome approach. Due to the limitations of separation method and mass spectrometer in previous

studies, most identified proteins in previous reports were high or medium ones, therefore, these proteins had more chances to be identified in our study.

## The functional annotation of the WAT proteome

To explore the function of WAT more comprehensively, PANTHER classification system (http://www.pantherdb.org/genes/batchIdSearch.jsp) and IPA tool (http://www.ingenuity.com/) were applied to the functional classification and pathway analysis.

The identified proteins were functionally categorized using GO annotation terms. As shown in Fig. 3A, nearly 85% of the proteins were present in cells, including cell part (40.6%, including intracellular part and endomembrane system), organelles (26.3%, including intracellular organelle and non-membrane bounded organelle) and macromolecular complex (17.4%, including membrane protein complex, catalytic complex, and ribonucleoprotein complex). The major molecular functions of WAT proteins were catalytic activity (44%, including hydrolase activity and transferase activity) and binding (37.5%, including protein binding and heterocyclic compound biding). High-abundance proteins were enriched with antioxidant activity (5.9%) and structural molecular activity (10.1%) when compared with the middle and low-abundance proteins (Fig. 3B). Using iBAQ analysis, the high-abundance proteins with antioxidant and catalytic activity accounted for 35% of all the high-abundance proteins, and the abundance of proteins with structural molecular activity accounted for 8%. For biological processes (Fig. 3C), approximately half of the WAT proteins were associated with metabolic processes (25.1%) and cellular processes (31%). High-abundance proteins were enriched with cellular component organization or biogenesis (11.3%), multicellular organismal process (4.5%), and biological adhesion functions (1.7%). The above results show that energy metabolism is an important function of WAT.

To identify the major biological pathways involved with the identified proteins, IPA was used for canonical pathway enrichment analysis (Fig. 3D; Fig. S5; and Table S5). The functions of WAT include protein synthesis (such as the protein ubiquitination pathway, EIF2 signaling, mTOR signaling, and the regulation of eIF4 and p70S6K signaling), energy metabolism (such as mitochondrial dysfunction, sirtuin signaling, PI3K/AKT signaling, and oxidative phosphorylation), adipogenesis (such as PI3K/AKT signaling), and the regulation of insulin action (such as integrin signaling). The function of the high-abundance proteins was mainly related to energy metabolism (mitochondrial dysfunction, oxidative phosphorylation, and sirtuin signaling). The primary function of the middle-abundance proteins was protein synthesis (mTOR signaling, regulation of eIF4 and p70S6K signaling, EIF2 signaling and tRNA charging). The functions of low-abundance proteins were WAT beiging (NGF and IL-4 signaling) and lipid metabolism (super pathway of inositol phosphate compounds, 3-phosphoinositide degradation, and 3-phosphoinositide biosynthesis).

## The analysis of WAT secreted proteins

Endocrine function is one of the key roles of adipocytes, and adipose tissue can synthesis several hormones, called adipokines or adipocytokines. We applied the SignalP 4.1 Server
A

B

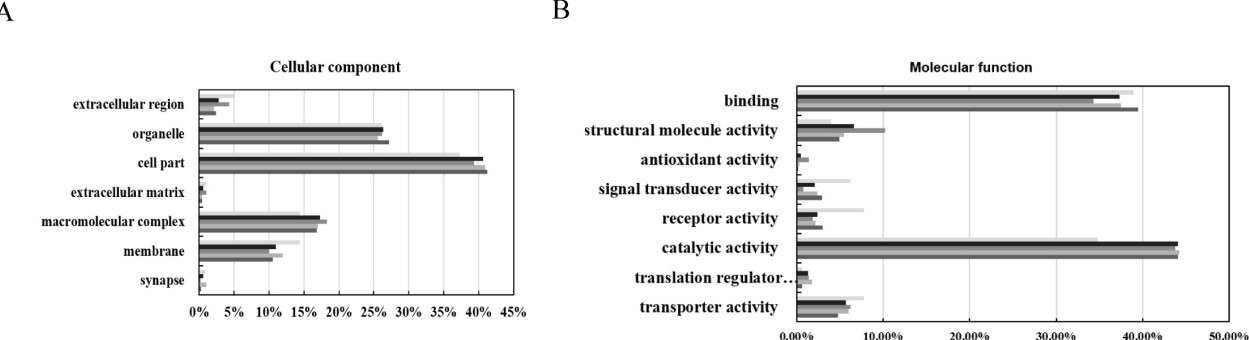

C

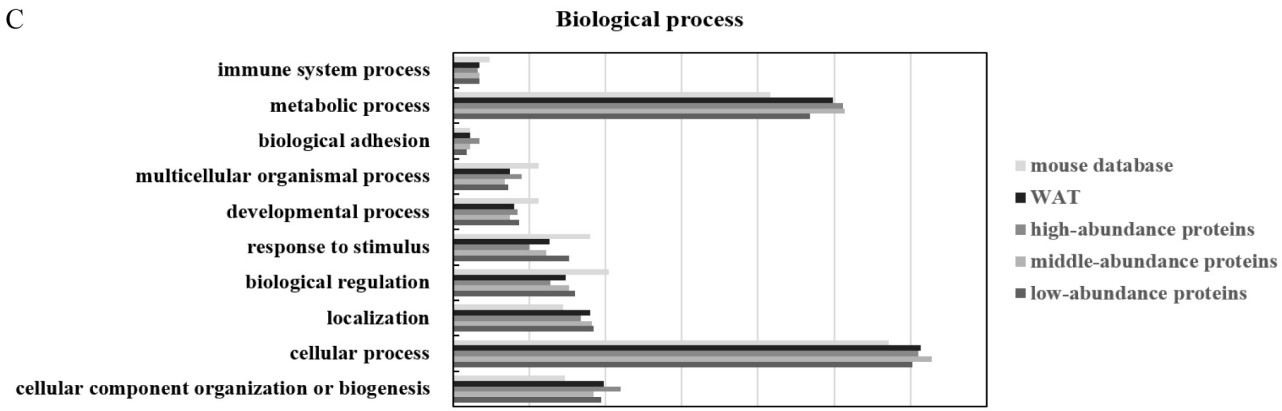

D Canonical pathway of WAT

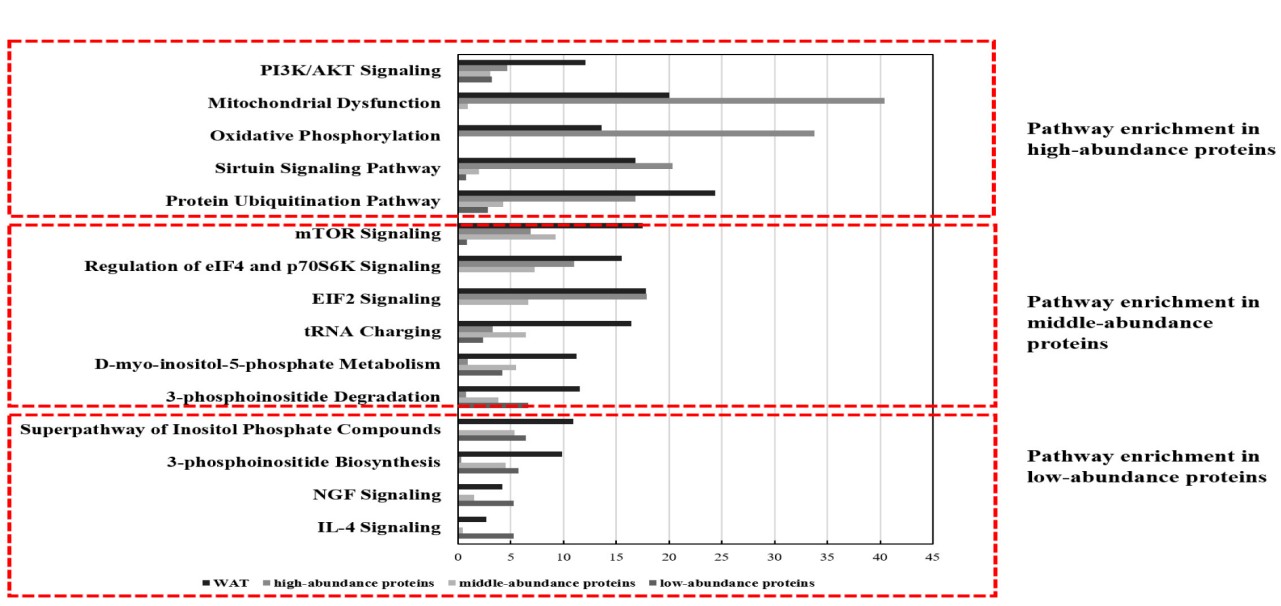

**Figure 3 Functional annotation (GO and IPA analysis) of WAT proteome.** (A) Cellular components, (B) molecular function, (C) biological processes, and (D) top canonical pathway.

to predict potentially secreted proteins. In all, 800 proteins were found to have signal peptides (Table S6), of which 200 proteins belong to the high-abundance proteins, 312 belong to the middle-abundance proteins, and 183 belong to the low-abundance proteins. The abundance of all the proteins with signal peptides was 15.6% in the WAT proteome. Ali Khan et al. (2018) identified 453 proteins from adipocytes that had previously been reported to be secreted proteins, of which 265 proteins were found in our results. Moreover, five WAT proteins were confirmed as adipokines: resistan, ADI, apolipoprotein E, haptoglobin, and angiotensinogen (Trayhurn & Beattie, 2001; Trayhurn & Wood, 2004; Yue et al., 2004).

In addition, the peak intensity-based semiquantification method iBAQ was applied to estimate the abundance of the identified 800 proteins, of which 694 proteins could be quantified (Geiger et al., 2012). The relative protein amounts span five orders of magnitude, from Semaphorin-4A (9.74E-01) to alpha-2-HS-glycoprotein (3.81E+04) (Fig. 2C).

## DISCUSSION

White adipose tissue stores energy in lipid droplets in the form of TAGs (Kuhnlein, 2012) and secrete hormones and adipokines to regulate energy intake, energy expenditure, and carbohydrate and lipid metabolism (McGown, Birerdinc & Younossi, 2014). A comprehensive profile of the WAT proteome will provide a basic understanding of WAT function and benefit disease studies. In the present study, we established a large database for the WAT proteome by advanced 2D LC–MS/MS. Total 6,039 proteins were identified, of which 5,270 proteins could be quantified by the peak intensity-based semi-quantification method iBAQ. Functional annotation was further explored and we found that proteins of different abundance performed different functions.

Canonical pathway enrichment analysis revealed that one of the primary functions of WAT was protein synthesis, and related significant pathways were EIF2 signaling. Global translation and apoptosis can be started by phosphorylation of EIF2-a (Kimball, 1999) and eIF4E was released by phosphorylation of eIF4E-BP binds to 5′ capped mRNAs to enhance their translation (Swetha & Ramaiah, 2015). Another significant WAT pathway is the PI3K/AKT signaling pathway, which plays a significant role in phytol-enhanced adipogenesis and glucose uptake of adipocytes and lipolysis inhibition (Langin, 2006; Williams, Kang & Wasserman, 2015). These results show the synthetic metabolism function of WAT, which is consistent with the traditional energy storage role of WAT (Trayhurn & Beattie, 2001). Furthermore, a previous study has suggested that adipose tissue plays an important role in insulin resistance (Chen & Hess, 2008). In our result, integrin signaling performed this function. A total of 103 proteins were found to be involved in this pathway, such as Akt and ILK. ILK can interact with the integrin cytoplasmic domains, and integrins rely on ILK for signal transduction. The pseudokinase domain of ILK is an important domain for the recruitment of adaptor proteins and/or signaling molecules, such as Akt and PDK1. Integrins are important regulators of insulin action and represent novel therapeutic targets to treat the underlying insulin resistance associated with T2D (Williams, Kang & Wasserman, 2015).

The WAT proteome was divided into high, middle, and low abundance proteins by the iBAQ quantitative method. According to IPA functional annotation, high-abundance proteins were mainly related to energy metabolism (mitochondrial dysfunction, oxidative phosphorylation, and sirtuin signaling). Sixty-five proteins were involved in sirtuin signaling, such as sirt2. During energy deficiency (caloric restriction or fasting) the expression of sirt2 increases (*Wang & Tong, 2009*; *Wang et al., 2007*). In this situation, adipogenesis was suppressed and adipocyte lipolysis was increased to promote fuel availability by sirt2 (*Wang & Tong, 2009*). In another significant pathway, oxidative phosphorylation occurs in mitochondria and comprises an electron-transfer chain, which is driven by substrate oxidation and is coupled to the synthesis of ATP through an electrochemical transmembrane gradient. When energy needs increase, fatty acid oxidation supplies an important source of energy for ATP production (*Boudina & Graham, 2014*). In oxidative phosphorylation, 60% ubiquinone oxidoreductase (including NDUFS4, NDUFA3, NDUFA4), 46% ubiquinol cytochrome c oxidoreductase (including UQCRFS, UQCRC2), and 63% ATP synthase (including ATP5A1, ATP5D) were high-abundance proteins (Fig. S6A). The other function of mitochondria is adipogenesis, which is accompanied by mitochondrial biogenesis (*Choo et al., 2006*). The mitochondria provide key intermediates for the synthesis of TAG might and further plays a critical role in lipogenesis (*De Pauw et al., 2009*). For mitochondrial dysfunction, many significant proteins were high-abundance proteins, including aconitase2 and cytochrome c oxidase. Aconitase2 was an enzyme to catalyze the interconversion of citrate via cis-aconitase in the second step of the TCA cycle. Cytochrome c oxidase was the terminal enzyme of the mitochondrial respiratory chain and a multi-submit enzyme complex to transfer electrons in mitochondrial (Fig. S6B). These results showed that besides basic physiological functions (structural molecular functions), high-abundance proteins also have important energy metabolism functions in WAT.

Ingenuity pathway analysis analysis showed that the primary function of the middle-abundance proteins was protein synthesis (mTOR signaling, regulation of eIF4 and p70S6K signaling, EIF2 signaling and tRNA charging). According to a previous study, protein synthesis is associated with adipogenesis (*Wilson-Fritch et al., 2003*). The concentration of numerous mitochondrial proteins such as cytochrome c had a 20- to 30-fold increase during differentiation of 3T3-L1 cells (a preadipocyte cell line). The expression of many nucleus-encoded mitochondrial genes during adipogenesis was also found to have statistically significant increases by mRNA profiles. The increase in mitochondrial proteins is not completely achieved by enhanced transcription but is also involved in post-transcriptional regulation. Adipose conversion (preadipocytes to adipocytes) would be accompanied by mitochondrial proliferation, which underscores the essential role of mitochondria in the key aspects of lipid metabolism (*Wilson-Fritch et al., 2003*).

The functions of low-abundance proteins were WAT beiging (NGF and IL-4 signaling) and lipid metabolism (super pathway of inositol phosphate compounds, 3-phosphoinositide degradation, and 3-phosphoinositide biosynthesis). NGF signaling plays a key role in the beiging of WAT by regulating the plasticity of intra-adipose

sympathetic arborizations in mouse inguinal WAT in response to a cold challenge. Intra-adipose sympathetic plasticity was suppressed by the blockage of NGF signaling, and moreover, the cold-induced beiging process of WAT was suppressed (*Cao, Wang & Zeng, 2018*). IL-4 signaling is involved in the cold-induced biogenesis of beige fat. All of 22 proteins were found to be involved in this pathway, including STAT6. STAT6 is thought to be necessary for the cold-induced remodeling of WAT into beige fat. *Qiu et al. (2014)* found that cold-induced expression of the protein UCP1 in subcutaneous WAT decreased by ~12-fold in Stat6−/− mice, accompanied by an increase in oxygen consumption. Accordingly, Stat6−/− mice maintained thermal homeostasis at a lower body temperature. Because beiging of WAT results in enhanced energy expenditure, it has increasingly gained attention for its potential application in the prevention and treatment of obesity and type 2 diabetes (*Giordano, Frontini & Cinti, 2016*; *Harms & Seale, 2013*).

In addition, 800 proteins were found to be secreted proteins in our result, of which 694 proteins could be quantified by iBAQ. The top five newly identified (most abundant) secreted proteins were carboxylesterase 1D, serine protease inhibitor A3K, apolipoprotein A-I, carboxylesterase 1C, and fibrillin-1. Carboxylesterase 1D, a major lipase in WAT, may mediate some or all hormone-sensitive lipase-independent lipolysis in adipocytes (*Soni et al., 2004*). Apolipoprotein A-I was associated with cholesterol transport in high-density lipoprotein (*Sackmann-Sala et al., 2014*). Fibrillin-1 was related with tissue homeostasis (*Pereira et al., 1999*). It was reported to be up-regulated in the WAT of diet-induced obese mice (*López et al., 2004*), which was affected by the structural changes when adipose tissue enlargement. The other two proteins were serine protease inhibitor A3K and carboxylesterase 1C. Serine protease inhibitor A3K was a contrapsin to inhibit trypsin-like proteases (*Takahara & Sinohara, 1982*; *Takahara & Sinohara, 1983*). Carboxylesterase 1C was involved in the detoxification of xenobiotics and in the activation of ester and amide prodrugs and in the extracellular metabolism of lung surfactant. No studies were reported to explore the correlation between them and WAT. Their functions in WAT need more work.

## CONCLUSION

In conclusion, using hp-RPLC integrated with a TripleTOF 5600 mass spectrometer, we provided the largest dataset from the normal mouse WAT proteome up to now. Our WAT proteome included many proteins that had not been previously reported by proteomic approach. We further provided insight into WAT function based on protein functional annotation and found that proteins of different abundance performed different functions. Finally, potentially secreted proteins prediction was performed. Our data set provided a comprehensive mouse WAT proteome, which might be useful for WAT function research.

In the future, the following issues should be addressed: With the emergence of new instruments (e.g., Orbitrap Fusion Lumos Tribrid mass spectrometer), labeling reagents (e.g., Ten-plex TMT labeling), and new techniques (e.g., the Data Independent Acquisition technique), high-throughput proteome analysis has come to fruition and might provide a more precise quantification and functional analysis. Besides, multiple sample preparation methods and adipose tissue samples from multiple regions will be needed to truly

obtain a comprehensive WAT proteome. Furthermore, with the development of genomics, transcriptomes, proteomics, and metabolomics, the combination of multiple omic analysis would lead to a better platform for routine clinical research.

### Funding

This work was supported by the National Basic Research Program of China (Nos. 2013CB530805 and 2014CBA02005), the National Key Research and Development Program of China (Nos. 2016 YFC 1306300 and 2018YFC0910202), the Key Basic Research Program of the Ministry of Science and Technology of China (No. 2013FY114100), the National Natural Science Foundation of China (Nos. 30970650, 31200614, 31400669, 81371515, 81170665 and 81560121), the Beijing Natural Science Foundation (Nos. 7173264 and 7172076), the Beijing Cooperative Construction Project (No. 110651103), the Beijing Science Program for the Top Young (No. 2015000021223TD04), the Beijing Normal University (No. 11100704), the Peking Union Medical College Hospital (No. 2016-2.27), the CAMS Innovation Fund for Medical Sciences (2017-I2 M-1-009) and the Biological Medicine Information Center of China, National Scientific Data Sharing Platform for Population and Health. The funders had no role in study design, data collection and analysis, decision to publish, or preparation of the manuscript.

### Grant Disclosures

The following grant information was disclosed by the authors:
National Basic Research Program of China: 2013CB530805 and 2014CBA02005.
National Key Research and Development Program of China: 2016 YFC 1306300 and 2018YFC0910202.
Key Basic Research Program of the Ministry of Science and Technology of China: 2013FY114100.
National Natural Science Foundation of China: 30970650, 31200614, 31400669, 81371515, 81170665 and 81560121.
Beijing Natural Science Foundation: 7173264 and 7172076.
Beijing Cooperative Construction Project: 110651103.
Beijing Science Program for the Top Young: 2015000021223TD04.
Beijing Normal University: 11100704.
Peking Union Medical College Hospital: 2016-2.27.
CAMS Innovation Fund for Medical Sciences: 2017-I2 M-1-009.
Biological Medicine Information Center of China.
National Scientific Data Sharing Platform for Population and Health.

### Competing Interests

The authors declare that they have no competing interests.
## Author Contributions

- Xiaoyue Tang conceived and designed the experiments, performed the experiments, analyzed the data, prepared figures and/or tables.
- Juan Li conceived and designed the experiments, performed the experiments.
- Wei-gang Zhao performed the experiments, contributed reagents/materials/analysis tools.
- Haidan Sun contributed reagents/materials/analysis tools, authored or reviewed drafts of the paper.
- Zhengguang Guo contributed reagents/materials/analysis tools, authored or reviewed drafts of the paper.
- Li Jing performed the experiments, authored or reviewed drafts of the paper.
- Zhufang She performed the experiments.
- Tao Yuan performed the experiments.
- Shuai-nan Liu performed the experiments.
- Quan Liu performed the experiments.
- Yong Fu performed the experiments.
- Wei Sun analyzed the data, contributed reagents/materials/analysis tools, prepared figures and/or tables, authored or reviewed drafts of the paper, approved the final draft, adviced how to analyze the data.

## Animal Ethics

The following information was supplied relating to ethical approvals (i.e., approving body and any reference numbers):

All animals were handled according to the Standards for Laboratory Animals (GB14925-2001) and the Guideline on the Humane Treatment of Laboratory Animals (MOST 2006a) established by the People's Republic of China. The two guidelines were conducted in adherence to the regulations of the Ethics Committee of Peking Union Medical College Hospital Institutional Animal Care and Use Committee (IACUC) and all animal procedures were approved by the IACUC (approval number: SCXKBeijing-2009-0004).

## Data Availability

The raw data are available at iProX under project ID: IPX0001504000 (https://www.iprox.org//page/project.html?id=IPX0001504000).

## Supplemental Information

Supplemental information for this article can be found online at http://dx.doi.org/10.7717/peerj.7352#supplemental-information.

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
