# Peer review of "Comprehensive map and functional annotation of the mouse white adipose tissue proteome"

_PeerJ, doi:10.7717/peerj.7352_

## Round 0.1 · original submission · Major Revisions

Please carefully address all the critical points raised by both reviewers and revise manuscript accordingly.

Reviewer 1 ·

Basic reporting

The manuscript was written clearly, with a nice summary of contemporary studies in the introduction. It would be even more helpful if the authors could elaborate more on what makes the current study stands out from previous reports. Importantly, unbiased interpretation of MS data and an insightful discussion would be appreciated.

Article structure is clear, and raw data was provided. Figures 1 and 2 are clear. Figure3 needs revision. Secretome section needs to be expanded.

The results provided are relevant to the research theme, but are not strong enough to draw concrete conclusions. Ideally, further experiments (such as western blot) should be performed to validate their results.

Please also see my general comments below.

Experimental design

This manuscript falls within the scope of PeerJ.

It would be helpful if the authors could further explain what’s the deficiency of previous studies. The use of cultured cells instead of real tissue sample? Why could the authors detect more proteins? Better spectrometer and sensitivity so low abundance proteins could be detected? ……

The research question and hypothesis are not clearly defined. It is unclear what’s the goal of this study, to more precisely map the mouse WAT proteome by identifying more proteins? It would be much appreciated if the authors could perform more analysis and further discuss how their new dataset contributes to our current knowledge of the biology of WAT.

Methods section are clearly written.

Please also see my general comments below.

Validity of the findings

Have the authors examined for contaminations in WAT sample? Can the authors validate their WAT samples were indeed WAT by any experimental data?

As mentioned above, the manuscript is lacking meaningful discussion, and it is unclear how this study would benefit the field.

It’s hard to imagine one MS dataset could serve as a baseline reference for future studies, as tissue samples are known to be heterogeneous and require complex preparation procedures. The fact that the most abundant protein observed is serum albumin suggests that WAT samples were contaminated, maybe by blood cells (?). If that is the case, how could the authors confidently analyze data and conclude that the ‘secreted proteins’, particularly for low abundance proteins, are actually synthesized in WAT other than being transported from elsewhere by blood?

Please also see my general comments below.

Additional comments

Obesity and diabetes are important topics in the biological and medical fields. Proteome studies of tissue samples are demanding to reveal the physiological characteristics of specific tissues and the pathological changes. The authors performed proteomic profiling of interscapular WAT collected from female mice, identified >6000 proteins covering about half of the expected human proteins, and predicted 800 secreted proteins of WAT. Aside from the comments above, below are some more detailed comments:
1. The authors reported 3500 newly identified proteins, are they less abundant than the known WAT proteins from other studies? Maybe add a panel in figure 2 showing the distribution of protein abundance? It would be appreciated if the authors could comment on why their study shared only a fraction (50%-60%) of proteins with previous reports. Please make a similar fig2d but only use the most abundant proteins in this study?
2. Why did the authors choose to use interscapular WAT? Please include this information in figure1 or at least in figure legends. Could the authors comment on the difference between interscapular WAT and abdominal WAT (Xie et al., 2010)?
3. The identification criteria in this study are different than others listed in Table2, please explain the reasons.
4. Line 187, how are these 5270 proteins selected? This number did not appear in any figure or other parts of the text. Why not use the 5057 proteins identified in all 3 runs?
5. Why didn’t the authors comment on demin and transgelin, two of the 'ten most abundant proteins'? What are they and their relationship to WAT? Are they identified for the first time in WAT?
6. In figure 3a-c, no legend is provided. Five bars were shown in each panel, what are they? In figure 3d, it would be helpful if the authors could use dash lines to divide the figure and highlight pathways for high, medium, and low-abundance proteins respectively. Please use a larger font in the main figure.
7. Could the authors expand their discussion on the predicted secreted proteins? What are the top 5 or 10 newly identified (most abundant) secreted proteins? Are they related to the functions of WAT?
8. Typos and grammar:
-line 84: The mice were fed a standard chow (SC; 10% lipids) diet for 22 weeks.
-line 162: and low-abundance ptoteins were analyzed respectively
-line 217: The major molecular function of WAT proteins were …
-line 252: The function of the high-abundance proteins were mainly related to…

Reviewer 2 ·

Basic reporting

Although the manuscript is generally understandable, there are numerous instances where the statements are scientifically inaccurate:
68-69: ‘proteomics technology has garnered many attention recently in biological research’. Many attention is not accurate as attention is not a quantifiable term.
Experimental procedure: use subscripts where necessary, for example: NH4HCO3
Ln 162: typo in ‘ptoteins’
Ln 170-171: In this study, samples of WAT from six C57BL/6J female mice were applied to generate a comprehensive profile of the normal mouse WAT proteome.
‘applied’ is not a proper term here to describe what scientific procedures were performed.
‘The relative protein abundance spans 6 orders of magnitude (Fig. 2C). Additionally, Table 1 showed the ten most abundant proteins in the WAT proteome.’
‘showed’ is not the appropriate verb here – ‘lists’ might be a better term.
Additionally, this sentence can be misleading. The table lists most abundant proteins found in your samples of the WAT proteome and not the WAT proteome in general. Also, this is an observation and not a claim.

The authors have covered some literature in the introduction and results section but have missed several important related studies (see below). Purposeful or unintentional, in either case it is unacceptable.

Figures are hard to read, Figure 3, all panels for example. Please use legible font size. Easy reorganization of panels A to D in figure 3 should easily allow for large font size.

Authors should consider adding pathway maps/figures for results section involving the role of proteins in various cellular pathways.

Complex data is not well represented. Results would be easier to understand if authors provide graphical representation of proteomic distribution.

Since this is not a novel workflow, figure 1 is not informative and necessary.

215-217: ‘nearly 85% of the proteins were present in cells (40.6% proteins present in cell parts, 26.3% in organelles and 17.4% in the macromolecular complex).’ This statement is not clear. What is meant by ‘cell parts’ and which assemblies are referred to in ‘macromolecular complexes’?
218: again clarification is needed: The major molecular function of WAT proteins were catalytic activity (44%) and binding (37.5%).'

Experimental design

There are serious flaws with the study design and result interpretation. In general, this is incomplete study with inconclusive results. The strength of this paper is a combination of technique and data analysis workflow. Label-free 2DLC/MS/MS is a valuable technique for proteomic analysis and when combined with the quantification algorithms such as iBAQ and bioinformatics workflow in a way that authors have used can lead to insightful results. But authors (1) have not designed a study with clear hypothesis, (2) have not optimized their sample preparation protocol and (3) have not utilized this powerful workflow to gain insightful results. Assuming that the aim was to develop a comprehensive murine WAT proteome reference dataset, the hypothesis is that the workflow that authors have used leads to the most comprehensive murine WAT proteome profile. However, authors have neither analyzed their proteome coverage nor have tried to optimize the coverage. With more than 10% of the most abundant proteins being serum albumin, the sample preparation methods is flawed. There are several studies that have reported difficulties in isolating WAT/BAT proteome stemming from high lipid contents in these samples. In the experimental process, the authors centrifuge samples following homogenization, and harvest supernatant for proteomic analysis. Did authors not observe a later of fat/lipids at the top following the centrifugation step? Generally, several rounds of extractions are needed to clarify fat and get the ideal protein content, which is generally a very small fraction in WAT samples. Several commercially available kits are used particularly to get around the problem of high fat and low protein contents in WAT samples.

Authors bin the proteins in 3 groups based on their abundance, no where in the study do author attempt to optimize the coverage for low abundance protein. If the aim was to comprehensively cover the WAT proteome, authors need to address the flaws in sample preparation and optimize good coverage for low abundance proteins to complete this study. Furthermore, authors suggest that removing albumin would be a good next step in their paper. It is not the next step, it is something absolutely needed to be done as a part of the current study to see some useful data.
195-200: authors mention very briefly the metabolic process/pathway associated with high abundance protein. A lot more efforts are needed to carefully analyze data to see if protein abundance correlates with dominant metabolic processes in WAT and if so, can other associated proteins be identified?
201-206: author compare proteome coverage in the current study with 5 other papers. Authors claim to have identified 3500 proteins that have not been identified before. Assuming that improved proteome coverage reflects improvement over other studies, how do authors explain missing 1061 protein from Forner et al study 1082 proteins from Adachi study and make a claim that current proteome reporting provides comprehensive coverage. Further efforts and the use of multiple sample preparation methods are needed to truly obtain a comprehensive WAT proteome.

The author claim to have identified new protein that were not covered in proteome datasets before, which is not a valid claim as explained below, but in its current state what additional information does this study provide that could not have been predicted from genomic or transcriptomic data sets?
Which anatomical locations were used to collect WAT samples?

Validity of the findings

Considering albumin was the most abundant protein, accounting for the 10% of total proteome – the WAT proteome samples are not representative of the actual physiological condition. Furthermore, the authors have tried to correlate observed protein abundance to cellular function, which, since the samples do not represent normal physiological protein distribution, are useless.

Authors compare the number of proteins identified with those identified in other studies, how was this comparison done?

Discuss protein size distribution

The 5 references the authors have used to compare their proteome with are from several years ago (2001-2009) and in several cases are the studies doe using human samples. Authors have failed to acknowledge and consider data from recent studies done on murine WAT and BAT including:
Feist et. al (2017) Quantitative proteomic analysis of murine white adipose tissue for peritoneal cancer metastasis. Analytical and bioanalytical chemistry, 410(5), 1583-1594.
Hsieh et. al (2013) Altered White Adipose Tissue Protein Profile in C57BL/6J Mice Displaying Delipidative, Inflammatory, and Browning Characteristics after Bitter Melon Seed Oil Treatment. PLoS ONE
Andrade et al. (2014) Proteomic white adipose tissue analysis of obese mice fed with a high-fat diet and treated with oral angiotensin-(1–7). Peptides
Kim et. al. Differential Protein Expression in White Adipose Tissue from Obesity-Prone and Obesity-Resistant Mice in Response to High Fat Diet and Anti-Obesity Herbal Medicines
Peinado et al. Proteomic Profiling of Adipose Tissue from Zmpste24−/− Mice, a Model of Lipodystrophy and Premature Aging, Reveals Major Changes in Mitochondrial Function and Vimentin Processing
Feist et al. 2018 Quantitative proteomic analysis of murine white adipose tissue for peritoneal cancer metastasis.
Ke et al. Differential proteomic analysis of white adipose tissues from T2D KKAy mice by LC‐ESI‐QTOF
Sackmann-Sala et al. Heterogeneity among white adipose tissue depots in male C57BL/6J mice.

---

## Round 0.2 · accepted · Accept

All critiques were addressed and the manuscript was amended accordingly. It can be accepted in its present form.

Reviewer 1 ·

Basic reporting

no comment

Experimental design

no comment

Validity of the findings

no comment

Additional comments

The authors have addressed my concerns.